# Position: You Can't Manufacture a NeRF

**MA Kimmel** [1]  **Mueed Rehman** [2]  **Yonatan Bisk** [3]  **Gary K. Fedder** [1]

## Abstract

In this paper, we examine the manufacturability gap in state-of-the-art generative models for 3D object representations. Many models for generating 3D assets focus on rendering virtual content and do not consider the constraints of real-world manufacturing, such as milling, casting, or injection molding. We demonstrate that existing generative models for computer-aided design representation do not generalize outside of their training datasets or to unmodified real, human-created objects. We identify limitations with the current approaches, including missing manufacturing-readable semantics, the inability to decompose complex shapes into parameterized segments appropriate for computer-aided manufacturing, and a lack of appropriate scoring metrics to assess the generated output versus the true reconstruction. The academic community could greatly impact real-world manufacturing by rallying around pathways to solve these challenges. We offer revised, more realistic datasets and baseline benchmarks as a step in targeting the challenge. In evaluating these datasets, we find that existing models are severely overfit to simpler data.

## 1. Introduction

Consider the shape in Figure 1. How would one create this shape as an output of a generative AI model? Consider the input and output modalities. Could you describe the shape with sufficient detail in text as input? What level of precision would be required to retain the filleted edges and smooth curves if a discrete output, such as a mesh or point cloud, were generated? Is there any way to create the clean surface segmentation seen in the leftmost image? How would this

[1]Department of Electrical and Computer Engineering, Carnegie Mellon University, Pittsburgh PA, USA [2]Electrical and Computer Engineering, Cornell University [3]Language Technologies Institute, Carnegie Mellon University, Pittsburgh PA, USA. Correspondence to: Ani Kimmel <akimmel@andrew.cmu.edu>.

*Proceedings of the $42^{nd}$ International Conference on Machine Learning*, Vancouver, Canada. PMLR 267, 2025. Copyright 2025 by the author(s).

shape be created using modern manufacturing processes?

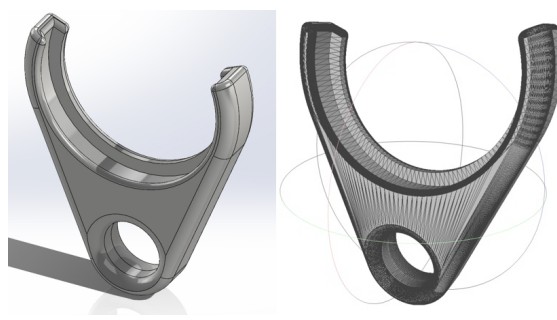

*Figure 1.* Example of a shape decomposed into its parametric boundary form on the left, discretely triangulated on the right

Modern physical environments are replete with a wide range of complex shapes and assemblies, most often realized through mass-manufacturing processes such as subtractive machining, casting, injection molding, or forming. Traditionally, these objects are designed by humans via a sequence of operations specified in computer-aided design software, which outputs a parametric boundary representation of the shape. Humans have the ability to both construct and decompose an object into logical sections or segments, even if sections do not have a nominal "classification" or semantic function. The shape in Figure 1 has clearly defined geometric segments as indicated on the left. It loses semantic context and precision when represented by a collection of triangles stitched together, as seen on the right.

From a mathematical standpoint, these shapes are necessarily "2-manifold watertight" objects (White, 2001), meaning that if each individual shape was represented as a polygonal mesh, every edge would be incident to exactly two faces. In addition to these raw physical constraints of the real world, in practice *parametrized*, *continuous-boundary* models allow for ease of editing, simulation (due to differentiability and interpolation) (Szabó & Babuška, 2021), and automated manufacturing (often involving feature-based machining) (Nasir & Sassani, 2021). Often formatted as either STEP or BREP files, these geometries usually bear additional useful features such as perfect symmetry, perfectly parallel or orthogonal surface normals, and completely consistent curvature (no bumps i.e. variability in surface normal rate-of-change) on a given surface. In contrast, discrete primitive

representations such as point clouds, voxels, and meshes, are usually suitable only for digital rendering and basic 3D printing given they have none of these desirable properties. For mass manufacturing, usually continuous equation-level 'perfect' accuracy is expected from CAD – errors will inevitably occur in the manufacturing process, approximating a shape with straight line segments and triangles to begin with would add an even greater level of error.

However, generating parametric, continuous-boundary representations natively out of generative AI models (that is, either BREP or CAD sequence generation) remains a challenge, and converting models into these forms is also nontrivial. Consequently, state-of-the-art (SOTA) 3D generation methods can be broadly categorized into two main classes (more strictly defined in the next section 1.1):

1. Models capable of generating *complex* objects in terms of discrete primitives (including neural radiance fields, point clouds, meshes, and voxels).

2. Models capable of generating *simple* objects in parametric boundary form, which to date have struggled with producing content that represents anything more than *simple* geometries. This is often called primitive-to-CAD generation.

We evaluate both types of models in this paper against a more realistic dataset and show the limitations of both.

What is desired for real-world manufacturing, is the ability to generate *complex* objects in parametric boundary form. Existing generative models do not train on datasets that have such features (as discussed further on) and do not generalize well to them. We have identified a common pattern in these generative strategies that prevents more effective generalization – a restrictive featurization step early on that is heavily dependent on hard labeling – and propose an augmented segmentation dataset that emphasizes a diverse, fine-grained segmentation of geometric shapes beyond conventional semantic or geometric primitive segmentation.

### 1.1. Definitions and Constraints

We introduce definitions of terms as well as some important geometric and manufacturing requirements for creation of 3D shapes that exhibit CNC-based manufacturability.

**Continuous vs. Discrete Shape Representations** As stated above, most shapes that are eventually manufactured come from continuous representations with easily identifiable and editable features. A variety of file formats exist, including STEP and BREP. They contain high-level representations of shapes, either as pre-defined parametrized objects or as free-form equations that describe a surface.

```
#24 = CIRCLE('NONE',#5177,1.5);
#25 = DIRECTION('NONE',(0.0,-1.0,0.0));
#26 = FACE_OUTER_BOUND('NONE',#6369,.T.);
#27 = ORIENTED_EDGE('NONE',*,*,#8503,.F.);
              ...
#5177 = VERTEX_POINT('NONE',#396);
```

*Figure 2.* Example of a portion of a STEP file in ASCII form, with explicitly defined high-level features.

There are also many discrete representation formats for shapes, including meshes, point clouds, and voxels. Meshes, commonly saved as STL or OBJ files, are a collection of vertices and edges without any defined features and are evaluated in this work.

$$
\begin{array}{l}
\texttt{solid name} \\
\left\{
\begin{array}{l}
\quad\texttt{facet normal}\, n_i\, n_j\, n_k \\
\quad\texttt{outer loop} \\
\qquad\texttt{vertex}\, v_{1x}\, v_{1y}\, v_{1z} \\
\qquad\texttt{vertex}\, v_{2x}\, v_{2y}\, v_{2z} \\
\qquad\texttt{vertex}\, v_{3x}\, v_{3y}\, v_{3z} \\
\quad\texttt{endloop} \\
\quad\texttt{endfacet}
\end{array}
\right\}^{+} \\
\texttt{endsolid name}
\end{array}
$$

*Figure 3.* STL ASCII format, which is a collection of triangular faces (facets) and vertices. $\{...\}^{+}$ indicates that the content is repeated for each facet. Values in italics are single precision floats, with positive vertices.

SOTA generation models for CAD reconstruction tasks take discrete representations of shape as input. To evaluate the performance of such models, it is crucial to derive the STL files (and from there, sample point clouds and their normals from generated meshes) from a dataset of parametric boundary files (e.g., STEP) to measure the reconstruction fidelity relative to the ground truth.

### 1.2. Classes of Objects

For the purposes of this paper, we shall define *complex* objects along the following criteria and *simple* objects as objects that do not match one of the following criteria.

A shape can generally be absolutely defined in one of three ways: its raw geometry, its construction sequence, or its machining sequence (including the tool path). We define *complex* objects as objects that satisfy any one condition (not necessarily all) of the following from three general but sometimes overlapping systems for defining a shape:

**1. Geometric properties** If the geometry in parametric boundary form contains more than 15 faces (the median of the unrestricted Thang3D dataset) or includes multiple spline-fit faces, it exhibits complex geometric properties. A spline-fit face is a surface in a 3D model defined using splines, allowing for smooth, freeform shapes rather than

simple geometric primitives like planes or cylinders. These faces are typically created using Bézier curves, B-splines, or NURBS, which interpolate between control points to form complex contours.

**2. Construction sequence**  When building a parametric shape, longer sequences—those exceeding 10 steps or involving more than seven extrude operations—tend to align more closely with realistic manufacturing processes. Complexity increases when the construction sequence extends beyond basic sketch-extrude-union-intersect operations to include advanced functions such as lofting, sweeps, skews, twists, mirrors, and CAD-specific tools like threaded hole creators, fillets, patterning, or coil creators.

**3. Machining sequence**  Machining complexity increases when an object contains machinable features such as bevels, fillets, chamfers, revolutions, or threading. Additional complexity arises when multiple non-planar machining features are present, or when the object requires multiple machines or multiple tool-head changes during production.

## 1.3. Requirements For Manufacturing Processes

Geometrically speaking, a boundary-less 2-manifold is a topological space $M$ whose points all have open disks homeomorphic to $R^2$ as neighborhoods. Colloquially known as "proper meshes" or "watertight meshes," this indicates that in a small locality anywhere on the mesh, the mesh is guaranteed to be planar. Strictly speaking, a 2-manifold mesh is a discrete approximation of a smooth surface where every vertex's local neighborhood—often called its "star"—is topologically equivalent to an open disk (and there are no half-disks since there are no boundaries in what we call valid meshes). Just as each point on a continuous 2-manifold locally resembles the Euclidean plane, each vertex in a manifold mesh must have incident faces arranged in a single, connected, cyclic order without gaps or overlaps.

Further clarifying 'watertight' in strict geometric terms:

1. A self-intersection is an intersection of two faces of the same mesh.
2. A non-manifold edge does not have exactly two incident faces.
3. The star of a vertex is the union of all its incident faces.
4. A non-manifold vertex is a vertex where the corresponding star is not connected when the vertex is removed.
5. A mesh is 2-manifold if it contains neither self-intersections, nor non-manifold edges, nor non-manifold vertices.
6. A 2-manifold mesh is watertight if each edge has exactly two incident faces, i.e., no boundary edges exist.

In practice, any holes, self-intersecting faces, or isolated faces, edges, and vertices violate these constraints and must be rectified before further processing can be applied (Deckner, 2024). This usually involves a threshold-based patching algorithm (Bernardini et al., 1999) that often requires human direction as there is no automated algorithm that can handle arbitrarily dense meshes. While some path-planning and slicing tools can handle small errors in the mesh, these kinds of errors will cause critical printing and structural stability issues even for custom 3D printing and especially at smaller layer heights (Montalti et al., 2024).

In subtractive manufacturing (e.g., CNC milling or turning) tight tolerances and advanced feature-based tool-path planning algorithms often dictate parametric inputs to the CAM software (Bianconi et al., 2006). Tolerance defines the permissible variation in a part's dimensions and geometry from its nominal design (Jensen, 2024), directly affecting how well parts fit together, their mechanical performance, and their durability under operational stress. Tight tolerances minimize defects such as misalignment or excess wear in assembled systems (Bode et al., 2022), reducing the need for post-processing and ensuring precise interactions between parts. In additive manufacturing, where layers of material are built up incrementally, tolerance control is equally crucial to avoid cumulative errors that could lead to weak points, poor surface finish, or dimensional inaccuracy. When tight tolerances are required, additive manufacturing is generally followed by subtractive finish machining.

Most mass-manufacturing operations, such as routing, turning, milling, engraving, screw machining, and metal casting, typically hold tolerances around ±0.005 inches (Ye, 2024). Cutting processes using specialized gasket tools or rail cutting tolerate around ±0.030 inches, while steel rule die cutting allows ±0.015 inches. Injection molding and laser powder-bed fusion both operate around ±0.1 mm (±0.0039 inches). Across these processes, a general purpose surface finish is commonly specified as an average roughness of 125 microinches (Ye, 2024).

As meshes scale in size, so too do their errors. For practical applications, it is crucial that they remain under an absolute tolerance for error regardless of the dimensions of the created object.

**Preservation of "Soft Edges"**  Bevels, chamfers, and fillets are common features across many manufactured shapes as shown in Figure 4. Both the generation and preservation of these "soft edges" are essential to real-world applications as these features offer critical mechanical and manufacturing advantages, often making important contributions to factors-of-safety and tool-path accessibility. True clean angle cuts are often difficult to achieve, while sharp angles meeting at joints concentrate stress and are not capable of

withstanding as much force as those with gradual transitions (Belingardi et al., 2002; Zielecki et al., 2017).

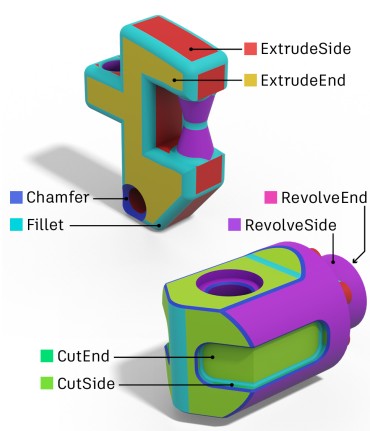

*Figure 4.* Examples of soft (fillet, chamfer, revolve) and hard (cut, extrude) features from the Fusion360 Segmentation Dataset as shown in (Lambourne et al., 2021).

### 1.4. Evaluation Datasets

With these constraints in mind, we sought a dataset that closely reflects real-world manufacturing scenarios by including both the construction sequence and the corresponding continuous boundary form (STEP or BREP). We derive mesh approximations of the STEP files in STL format to compare how accurately discrete representations capture the original geometry.

Most 3D CAD generation datasets have been intentionally restricted in ways that do not align with most real manufactured shapes. The DeepCAD Dataset (Wu et al., 2021) which was parsed from the ABC Dataset (Koch et al., 2019) intentionally only targets sketch-extrude sequences. MF-CAD++ (Colligan et al., 2022) is a synthetic dataset that was not created by humans, which tends to have a much simpler construction sequence involving single base extrusion with multiple subsequent "features" added. A recent dataset, CAD MLLM (Xu et al., 2024b), intentionally suppresses chamfer and fillet operations.

Our evaluation dataset is built upon the Fusion360 (F360) Gallery Segmentation Dataset of roughly 35,000 parts as it has models incorporating advanced construction features such as fillets and chamfers and is also designed by humans. Even so, some CAD operations were suppressed for simplification (Lambourne et al., 2021). The dataset includes corresponding STEP and STL representations of each part, though more precise STL meshes can be generated. We sample from these mesh approximations to generate point clouds along with associated surface normals, then label each point as a 'segment' according to their corresponding BREP face in the dataset. Notably, these labels do not correspond to any particular primitive class – they merely act

as segment clusters based on the BREP geometric split of the object. We posit that true BREP reconstruction requires this geometric level of segmentation. Additionally, we also evaluate on 200 models downloaded from the Thang3D online CAD file repository, which have no restrictions or simplifications.

## 2. Evaluation of Existing 3D Generation Work

We evaluate SOTA models in mesh generation as well as CAD reconstruction against the F360 segmentation dataset, and evaluate CAD reconstruction on the Thang3D dataset. We demonstrate that mesh alone, even at extremely high grid resolutions with noiseless inputs, is not precise or accurate enough for standard manufacturing techniques. We also demonstrate that SOTA CAD reconstruction similarly fails to reconstruct a mathematically valid object over 80% of the time on the complex datasets.

### 2.1. Limitations of SOTA Mesh & NeRF Generation

Recently, many 3D shape generation papers have appeared that generate objects either as point clouds, voxels, or meshes (Mittal et al., 2022; Siddiqui et al., 2024; Liu et al., 2023b; Lin et al., 2023; Wen et al., 2019; Xu et al., 2024a; Liu et al., 2024a). Still other works have delved deeper in text-to-3D-generation using NeRF or some form of it, including DreamFusion (Poole et al., 2022), ProlificDreamer (Zeng et al., 2023), Phidias (Wang et al., 2025), and others (Liu et al., 2023a; Zeng et al., 2023; Xie et al., 2024). These rendering-based approaches could, in theory, be used to generate meshes by surface meshing of the points. However, these models introduce variability, noise, and error even before the final output since they generate from encodings. Further, we demonstrate that even if these models could create meshes from precise inputs, the remeshing even from an infinitely fine-grained input discretization (e.g., a neural radiance field (NeRF) which offers a continuous implicit representation) would not satisfy the manufacturing requirements listed above. For the purpose of this paper, we evaluate meshes created from a neural radiance field under what we consider perfect conditions, namely:

1. The initial mesh being sampled is noiseless, exactly 2-manifold and water-tight.
2. The model can sample from as many "camera" positions and angles as necessary.
3. "Camera" angles are also known and exactly calculated.
4. No noise is added.

Using the F360 segmentation dataset, we allowed Instant-NGP (Müller et al., 2022) to train until either the total loss is less than 0.0025 or 250,000 steps are reached, with the results shown in Table 1. The output resolution for every mesh is 256 x 256 x 256, which is greater than the resolution

| Has Feature | # | % NM | Avg Chamf | NM Edges |
|---|---|---|---|---|
| Fillets | 7166 | 55 | 0.034±0.06 | 5.54±18.1 |
| Chamfers | 3045 | 60 | 0.029±0.05 | 6.76±12.7 |
| Revolve | 3975 | 69 | 0.036±0.07 | 6.45±10.5 |
| >7 Extrudes | 432 | 45 | 0.056±0.08 | 6.45±10.5 |
| All | 34871 | 50 | 0.029±0.07 | 6.67±22.1 |

*Table 1.* Comparison of shape metrics for NeRF generated from the F360 segmentation dataset, where all shapes are scaled to fit in the unit sphere. The first column specifies the feature in the subset, the second column gives the number of samples with that feature, the third column gives the percent of non-manifold (NM) generated meshes for objects with that feature, the fourth column gives the average chamfer distance and standard deviation between the original mesh and the NeRF output, and the final column gives the average number of non-manifold edges and standard deviation.

or maximum face number of any of the prior discrete generation models listed above. We remesh the output neural radiance field using an improved marching cubes algorithm, and all benchmarks are evaluated against these meshes.

**Evaluation** Even under these "optimal" circumstances, nearly 50% of these shapes contained non-manifold edges as seen in Table 1, with some having egregious errors. The reconstruction model especially struggled to accurate capture the objects that had revolutions, chamfers, or fillets. Objects with curves (fillets, chamfers, revolutions) and objects with many extrusions fared exceptionally poorly.

The median chamfer distance, though small as a percent error ranging from 1-3%, scales proportionally with the mesh. Some poor meshing approximations are shown in Figure 8, where one can observe choppiness in the mesh compared to the BREP files. For a 4 inch-long object, for example, the chamfer distance errors referenced in Table 1 would be an order of magnitude larger than that of the absolute tolerances listed earlier.

**Issues with Mesh Formatting** In a mesh representation, the magnitude of errors generally are too great to be considered valid for machining. Greater challenges persist in attempts to fix these errors. In the manufacturing space, STL files (often misleadingly thought of as valid 'meshes' despite a lack of adjacency information) are often known as "triangle soup" due to their unstructured, featureless nature as a collection of vertices and edges. As such, they are not capable of direct, targeted editing by CAD software and instead are patched via various thresholding algorithms including merging vertices, edges and faces, adding additional faces, unstitching negative gaps, and retriangulation – all based on human-chosen thresholds (Campen et al., 2012). While these algorithms might be able to eventually produce 2-manifold, watertight meshes, they will not be able to resolve general skewing errors on the mesh itself as illustrated in Figure 8. They also tend to produce situations where the

repair of one error tends to drive the creation of another error (Campen et al., 2012).

Minute changes in surface normals and curvature are hard to patch automatically, and smoothing algorithms must make qualitative trade offs between preserving sharp features and achieving a smooth surface. Even advanced methods like anisotropic, Taubin, or bilateral smoothing struggle with complex or fine meshes and volume preservation. Without defined contour lines or features, the mesh is barely editable beyond simple scaling, even if errors could be patched and surfaces were smoothed. With high face counts and strict tolerances, this format is not feasible for general manufacturing production.

### 2.2. Limitations of SOTA Primitive-to-CAD

Given these issues, greater focus by researchers has shifted to what is known as primitive-to-CAD reconstruction, where a CAD representation is generated from a mesh or point cloud input. Point2CAD (Liu et al., 2024b) introduces a pipeline for reconstructing CAD models from point clouds by segmenting the cloud into clusters and fitting geometric primitives or freeform surfaces using a novel neural representation. This approach focuses on a hybrid of neural and analytical methods to address topological consistency, thereby capturing edges and corners with a new level of accuracy. Point2Cyl (Uy et al., 2022) maps input points to extrusion cylinders in a construction sequence. Their predecessor, DeepCAD (Wu et al., 2021), introduced the first deep generative model based on transformer-encoder architectures that outputs 3D shapes as sequences of CAD operations, addressing the limitations of traditional generative models that rely on discrete representations like meshes and point clouds.

CAD-SIGNet (Khan et al., 2024) uses an auto-regressive neural network that reconstructs CAD design histories from point clouds, leveraging a multi-modal transformer architecture with cross-attention and a Sketch Instance Guided Attention (SGA) module for enhanced detail accuracy. Most recently, CAD-MLLM (Xu et al., 2024b) presents a multi-modal framework that generates parametric CAD models conditioned on various input modalities, including text, images, and point clouds. CAD-MLLM leverages a large multimodal dataset, Omni-CAD, and utilizes large language models to align these different modalities into a coherent CAD generation process. The approach outperforms prior models in CAD generation; however, it again only focuses on sketch-extrude-boolean sequences while explicitly removing chamfers and fillets so we hypothesize it will likely face the same challenges as Point2CAD.

All of the aforementioned generative models loosely follow a similar pipeline, shown in Figure 5, that is initially based on splitting a primitive form (meshes, points, voxels) into

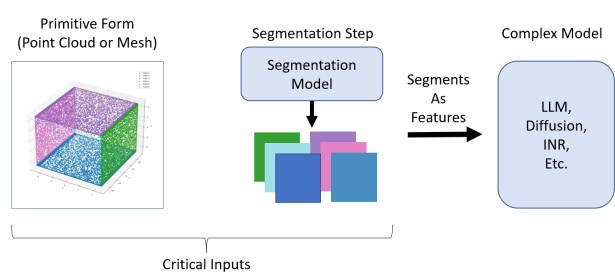

| Has Features | # of Samples | % NM | NM Edges |
|---|---|---|---|
| Fillets | 7166 | 94 | 3423±2813 |
| Chamfers | 3045 | 90 | 2783±1906 |
| Revolve | 3975 | 89 | 2224±1428 |
| > 7 Extrudes | 432 | 98 | 3267±2840 |
| **All** | 34871 | 81 | 1834±4194 |

*Table 3.* Metrics for Point2CAD output CAD files where the original shape is scaled to fit in the unit sphere. The first column specifies the feature in the subset, the second column gives the number of samples with that feature, the third column gives the percent of parts having meshes that are non-manifold (NM), and the final column gives the average number of non-manifold edges and standard deviation.

*Figure 5.* Common model pipeline for primitive-to-CAD reconstruction, where a primitive is broken down to create feature vectors according to pre-defined classes or labeled tokens.

| | % Manifold | | |
|---|---|---|---|
| **Model** | ABC* | Fusion 360 | Thang3D |
| **Point2CAD** | 92.1 | 18.9 | 6.9 |
| **Point2Cyl** | 81.2 | 12.3 | 0 |

*Table 2.* Comparison of Primitive-to-CAD reconstruction models across datasets, showing what percentage were manifold shapes. *ABC evaluation performed on a random subset of 10,000 shapes.

geometric segments or mapping an encoded primitive to a set of explicit CAD-related tokens. These subsets are then turned into features, which are fed into another model or models. The inability to properly extract features leads to major inaccuracies later in the pipelines because unrecognized segments cannot be matched to existing primitives.

**Evaluation** We test the Point2CAD and Point2Cyl models using both the F360 segmentation dataset and the Thang3D dataset. We densely sample 10,000-point point clouds and their corresponding surface normals from the mesh version of each file as input. Ground-truth segmentation is defined by labeling each point in the sampled point cloud according to its closest BREP face.

Results shown in Table 2 indicate that the segmentation step does not perform well on the F360 segmentation dataset or the Thang3D dataset. Over 80% of the reconstruction outputs of the F360 segmentation dataset and over 90% of the Thang3D dataset are non-manifold for Point2CAD. Furthermore, this table alone is not a true capture of the error, as further discussed in section 3. Intrinsically, any model that relies on a set number of discrete input points will be mathematically incapable of representing arbitrarily complex surfaces. The state of many of the output files is so distorted that it is of little value to attempt to compute chamfer distance between it and the ground truth objects, as exemplified in Figure 6.

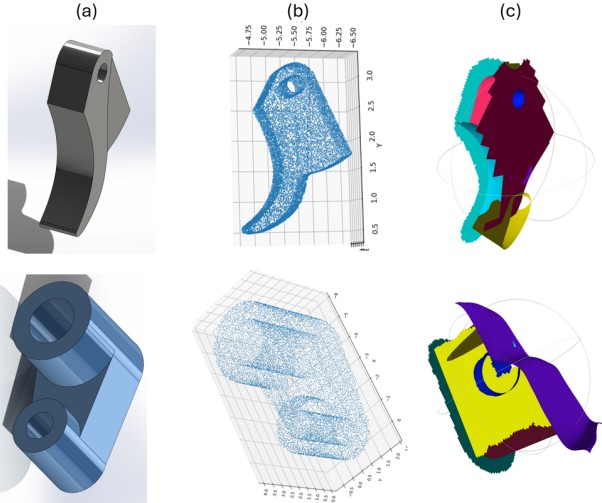

*Figure 6.* Examples of Point2CAD reconstruction errors where (a) shows the original shape, (b) shows the sampled point cloud, and (c) shows the attempted reconstruction.

## 3. Existing Challenges

**Classless Segmentation and Decomposition** The stitching together of primitives via boolean operations, as in constructive solid geometry, results in a much smaller subset of shapes than what is seen in the modern world today. Humans possess an innate ability to partition complex shapes by identifying coherent regions of continuous or similarly varying surface characteristics, such as curvature or normal direction, even when those regions do not conform to standard geometric primitives (e.g., planes, cylinders, or spheres). We can construct a wide variety of non-prismatic objects purely by deciding what geometric features we would like the object to contain as opposed to selecting from subset of classes and applying boolean operations to them. We are also capable of recognizing when there are multiple valid geometric decompositions for a shape, and when decompositions are invalid.

In contrast, nearly all reconstruction and generation, including all the work discussed in this paper, has been heavily predicated on featurization performed on the basis of a small vocabulary of defined classes, either geometrically or via their construction sequence (usually limited to sketch-extrude sequences, with some extra commands such as 'slot' or 'hole'). Inheriting from the RANSAC primitive-fitting algorithms (Schnabel et al., 2007) before them, these models are able to use simpler metrics such as Mean Intersection over Union (mIoU) to train since they act on common geometries associated with pre-defined labels. Whether it is assigning subsets of the primitives to a geometric shape (Point2CAD, ParseNet, BrepGen) or assigning subsets of primitives to CAD sequence instructions (CAD SigNet (Khan et al., 2024), DeepCAD, HierarchicalCAD (Colligan et al., 2022)), there is always a "label." The hard association of classes to primitive subsets as opposed to a more generic, parametric understanding of geometry is intrinsically limiting.

**Permutation Invariant Metrics**  Achieving classless segmentation and decomposition necessitates a loss function that does not actually weight based on what specific label is chosen for each section, but rather that the sections are properly segmented. However, achieving a fully differentiable, permutation-invariant loss function that is not based on a discrete matching algorithm is challenging.

mIoU is a popular metric for segmentation performance, but it has two notable drawbacks in practice. First, as with training loss, it is not permutation invariant in the context of instance-level segmentation; different ways of matching predicted segments to ground truth labels can produce different mIoU scores, which forces the use of a separate, typically non-differentiable matching step (e.g., the Hungarian algorithm) to align predictions with targets before the IoU can be computed. This lack of permutation invariance stems from the fact that IoU is evaluated pairwise and thus depends heavily on how instances are paired. Second, mIoU does not explicitly account for scenarios where the model fails to predict an entire segment altogether—if a ground truth segment does not have a corresponding prediction, it is not penalized in a straightforward way by the mIoU computation alone. These limitations motivate the use of more sophisticated metrics that can both handle one-to-one matching in a principled way and penalize missing or extra segments.

**Reconstruction to Generation – Component Mapping**
Since a limited number of components are being used as input features for generation models such as CADSigNet, BrepGen, and CAD-MLLM, anything beyond these segments will inherently be out of distribution, limiting the output space of existing generation models.

**Imprecise Noise in Training**  Chamfers and fillets that are a very small fraction of the total size of an object can have a great impact on its mechanical properties. However, a common issue across many segmentation, reconstruction and generation models including Point2CAD, SamPart3D (Yang et al., 2024), BrepGen and Point2Cyl is the introduction of noise to the input primitive based on a flat threshold relative to the overall size of the object (often scaled to be within the unit sphere). Added noise obscures finer features and introduces a mollification or blurring of segments that makes it challenging to differentiate if an edge or corner is intended to be truly rounded or sharp.

**Mechanically Valid Shapes**  A shape being geometrically valid does not mean it will be mechanically valid according to all requirements and constraints. Mechanical design (i.e., performed by humans) is largely based on how physical forces, along with other effects such as heat, will interact with the designed object. Designs are generally verified with physics-based simulation, such as finite-element analysis, and eventually validated in real-world testing. Knowledge of how these systems work is a key component in the mechanical design process. Embedding that knowledge, which is dependent on design requirements and constraints, as well as guaranteeing explainability, fault tolerance, and factors of safety into a neural network remains an open challenge. Some progress been made in physics-informed neural networks (Raissi et al., 2019). Ensuring that there is a possible machining sequence that will create the shape (e.g., tool-path accessibility) is yet another challenge.

**Input Modalities**  Most generative design has targeted either image-to-3D or text-to-image-to-3D. In contrast, mechanical design by humans is performed through visual 3D interfaces with actions that translate to mathematical surface descriptors. Text and/or images are not sufficiently precise to describe a shape even for human design. The creation by industry of domain-specific language (the BREP/STEP vocabulary and 'grammar' structure) and visual interfaces to generate the language files ensures that shape by design is specified such that all surfaces have perfect 1:1 mapping in 3-space. It remains to be seen whether any input modality with greater ambiguity could truly be used in an industry setting, or if there are better inputs that can be used alongside machine-learning techniques that are more quantitative, like inputs to a deterministic topology optimizer.

## 4. Shape Decomposition: Open Challenge

We recognize that many of the issues with the aforementioned pipeline models such as Point2CAD arise from the initial featurization step being restricted to a certain subset of classes, which only recognizes basic primitives or basic CAD sequences that have clearly defined labels. We intro-

duce the relabeled F360 segmentation dataset as well as the fully unrestricted Thang3D dataset to address this issue.

Using ground-truth segmentation according to the BREP face assignments (i.e., ground-truth decomposition), the reconstruction error decreases, shown by the reconstruction percentage dropping to 47% non-manifold for the F360 dataset — all without any retraining of downstream models. Though these numbers are still below a practical threshold, they underscore the importance of generating much finer-grained decomposition than what is typically achieved by current segmentation efforts (such as being able to segment a fillet from a connecting plane), then leveraging those pieces as a far more diverse set of input features. To reach human-level complexity in design and construction, 3D generation models must be able to decompose and featurize geometry at the granularity that a human would – that is, create the BREP faces. We call this process "geometric decomposition" to distinguish it from the coarser, more limited geometric segmentation common in previous work, which often has hard labels attached to each kind of segment.

### 4.1. ParSeNet Geometric Decomposition

We test the ParSeNet segmentation model (Sharma et al., 2020) naively from the checkpoint created by training on the ABC dataset as used in Point2CAD using advanced sampling. We calculate the curvature (change in normal between the faces) for each edge in the mesh and densely oversample around high curvature edges proportional to the length of the edge, with a minimum number of samples per edge being 10. All objects are normalized to be within the unit sphere and samples of points with their normals are taken from the mesh.

We define edge curvature as the angle, $\theta$, between normals on either face associated with the edge. We assume every mesh is 2-manifold and watertight, meaning every edge will have exactly 2 connected faces. We also define a sampling threshold $t$ around these high curvature edges as $t = 0.035 \, m_d$ where $m_d$ is the minimum dimension of the bounding box of the object.

We define high curvature edges as any edges where $\theta$ is greater than 0.3 radians and ensure at least 25% of all sampled points across the mesh come from locations within the threshold $t$ sample zone around these edges. This guarantees a significant number of points will come from areas with high curvatures. The remaining 75% of samples are evenly distributed across the mesh surface according to face area, with a minimum number of three samples per face. After all samples are calculated, 10,000 points are randomly selected from the total number of points sampled.

The results are summarized in Table 4. The high curvature oversampling marginally improves the mIoU metric for all

| Class | mmIoU | # M. Seg | # F. Segs |
|---|---|---|---|
| Fillets | 0.22 | 0.55 | 16.53 |
| Chamfers | 0.22 | 0.463 | 16.80 |
| Revolve | 0.23 | 0.467 | 11.07 |
| > 7 Extr. | 0.16 | 0.02 | 44.42 |
| All | 0.23 | 1.52 | 8.90 |
| Fillets (NN) | 0.54 | 0.15 | 14.71 |
| Chamfers (NN) | 0.53 | 0.09 | 15.05 |
| Revolve (NN) | 0.51 | 0.17 | 9.84 |
| > 7 Extr. (NN) | 0.42 | 0.01 | 44.1 |
| All (NN) | 0.62 | 0.42 | 8.47 |

*Table 4.* Comparison of decomposition metrics, testing ParSeNet. mmIoU is matched mean IOU, meaning the average IoU for matched segments, ignoring completely missed or false segments. # M. Seg is the average number of missing segments (i.e., segments not identified). # F. Seg is the average number of false segments (i.e., identified segments that do not exist in the ground truth BREP). The bottom half of the table are results from evaluation with no normals (NN) input, just points.

classes. We speculate that a greater number of input points or a continuous input representation would improve the quality of segmentation as it is challenging to accurately capture all features with just 10,000 points.

Notably, the model performed worse with surface normals introduced than without it, in contrast to the data provided by the original ParseNet paper. While out of distribution data is always challenging, we hypothesize the model is over-indexing on sharp changes in normals as a class delimiter as opposed to truly parametrizing according to the normal vectors' values or rate of change of the normal vectors.

## 5. Alternative Views

Perhaps we do not need classless decomposition, we simply need more classes. This is a decidedly algorithmic approach, as opposed to a human one. While increasing total numbers of classes and/or parametrizing them will likely improve performance in reconstructing and generating diverse shapes, it fundamentally does not address the issue that humans split and generate shapes along logical geometric boundaries – not based on which class is most likely to be the next predictable token or what split will guarantee decomposed segments will fit into existing labels. The intrinsic questions "does this split create some sort of geometric consistency within the segments?" and "does this split increase the likelihood of a specific class or token?" might be linked, but the latter is far less general than the former. Our goal is for generative models to develop an intuitive understanding of geometry. It should be entirely possible to create segments that have never been seen before and do not fit any one class (or alternatively fit multiple) based on the local geometry of a shape – humans do this regularly.

Others might challenge the necessity of strictly coupling generative AI with tightly parameterized models for mass-manufacturing. Some believe additive manufacturing, where flexible mesh or voxel-based representations can already suffice, reduce the urgency for painstaking parameterization. There is also the question of whether older CAD paradigms should remain a firm requirement, given that neural and implicit representations can capture high-fidelity geometry and might be converted downstream to machinable formats. Early-stage approximations are highlighted for their conceptual value, with critics suggesting that enforcing precise tolerances during ideation risks stifling creativity.

While it is true that additive manufacturing workflows, non-parametric neural representations, approximate early-stage models, and human-in-the-loop iterations can each serve various purposes in modern product development, these viewpoints do not diminish the overall necessity of bridging generative 3D models with the strict manufacturing constraints demanded by high-volume, precision-focused industries. Humans in the loop will always want an output that they can edit easily. Additive manufacturing remains a complement rather than a complete substitute for subtractive processes like CNC machining, especially for parts that must meet tight tolerances or require specific finish qualities. Even so, as discussed above, many additive algorithms will not tolerate current mesh errors in SOTA mesh-generation models. Neural representations and learned implicit functions, while powerful, still require clear pathways to produce parameterized outputs that standard CAM pipelines can interpret; relying on post-hoc conversion from implicit to explicit geometry can introduce new sources of error and there is currently no automated pipeline for doing such conversions with arbitrary complexity. Approximate or creative designs are undeniably useful for concept generation but still require heavy human involvement and ultimately could take more time recreating in CAD software than it would have been to start from scratch with expert knowledge.

## 6. Conclusion

We have critically evaluated the limitations of existing datasets and modeling approaches for segmentation and reconstruction of parametric boundary-defined 3D shapes. We emphasize the inadequacies of mesh and other discrete representations for the purposes of real-world creation and demonstrate that many existing pipeline-style models and their training datasets do not effectively generalize to real world parts, highlighting the need for datasets that encompass greater geometric and manufacturing complexity. We also show that a significant portion of the error from these sets of models occurs in the featurization step, which is incapable of breaking down shapes into sufficiently elementary parts for reconstruction. We introduce a revised version of

the F360 segmentation dataset that is labeled according to each shape's BREP decomposition in addition to a smaller labeled dataset of real parts from Thang3D in an attempt to encourage more geometric diversity within the CAD generation space as well as much finer-grained segmentation (referred to here as shape decomposition). Finally, we establish a baseline for shape decomposition, using the modified F360 segmentation dataset and ParseNet, that is more accurate to the requirements for creating real-world objects and brings primitive forms closer to their higher-level BREP counterparts, allowing for better training on downstream tasks.

These findings underscore the importance of dataset diversity and authenticity to the problem it is trying to solve as well as the critical role initial featurization plays in pipeline style models for 3D generation tasks. We hope this augmented dataset will spur further developments in advancing parametric segmentation and reconstruction techniques, paving the way for more robust solutions to address the challenges of bringing useful shapes off the screen and into the real world.

## 7. Future Work

Achieving BREP-level accuracy in shape decomposition is the first step in bridging the gap between current generated 3D shape outputs and manufacturability. Future work could include attempting segmentation with non-discrete forms such as implicit neural representations, encoded mesh patches, or other ways of processing arbitrarily complex shapes where a finite number of discrete points would be too large to efficiently train as input and adding other geometric features would allow the models to converge faster.

Once fine-grained decomposition is achieved, next steps would involve training any of the aforementioned downstream pipeline models to reconstruct complex shapes as parametric boundary objects with these new input segments as features. The final goal would be to generate these sort of parametric boundary representations directly from ML models for the complex shapes presented in the real world.

Looking even further into the future, generating shapes in parametric boundary representation with some knowledge of physical parameters, machining accessibility, and even stricter requirements similar to topology optimization where a parameter like mass or volume was minimized while factors-of-safety was still enforced, remains a fascinating open challenge.

## Acknowledgement

This work was funded by discretionary funds of the authors.

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

# A. Appendix

Edge Curvature Equation:

$$\theta = \cos^{-1}\left(\frac{\mathbf{n}_1 \cdot \mathbf{n}_2}{\|\mathbf{n}_1\| \, \|\mathbf{n}_2\|}\right).$$

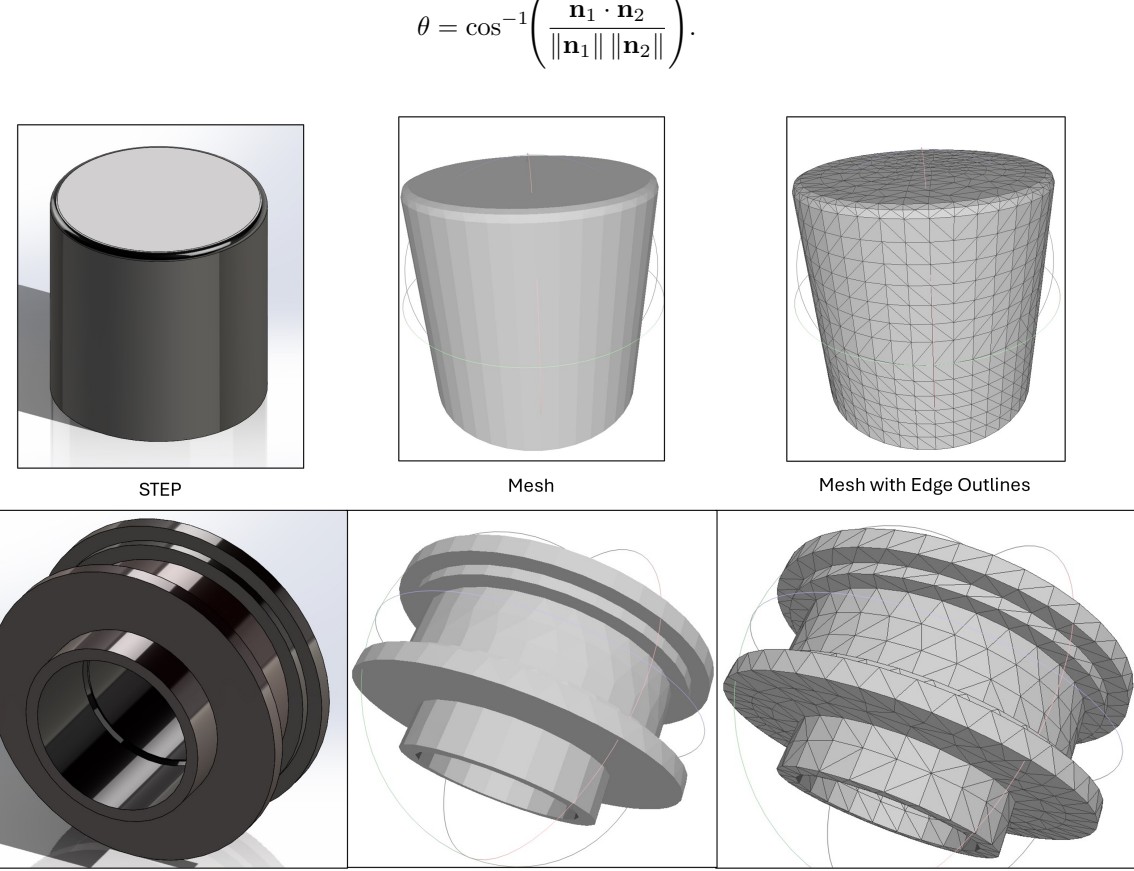

STEP     Mesh     Mesh with Edge Outlines

*Figure 7.* Examples of remeshed BREPs from the Fusion360 Segmentation Dataset with curvature precision errors (straight line segments do not approximate the shape within standard machining tolerances when scaled to 4 inches along the longest dimension).

| Process | Tolerance |
|---|---|
| Router | ± 0.005 in |
| Lathe | ± 0.005 in |
| Router (Gasket Cutting Tools) | ± 0.030 in |
| Milling (3-axis) | ± 0.005 in |
| Milling (5-axis) | ± 0.005 in |
| Engraving | ± 0.005 in |
| Rail Cutting Tolerances | ± 0.030 in |
| Screw Machining | ± 0.005 in |
| Steel Rule Die Cutting | ± 0.015 in |
| Injection Moulding | ± 0.1 mm (0.0039 in) |
| Metal Casting | ± 0.005 in |
| Laser Powder-Bed Fusion | ± 0.1 mm (0.0039 in) |
| Surface Finish | 125 RA |

*Table 5.* Standard Tolerances for Mass Manufacturing (Ye, 2024). RA is average roughness in microinches.

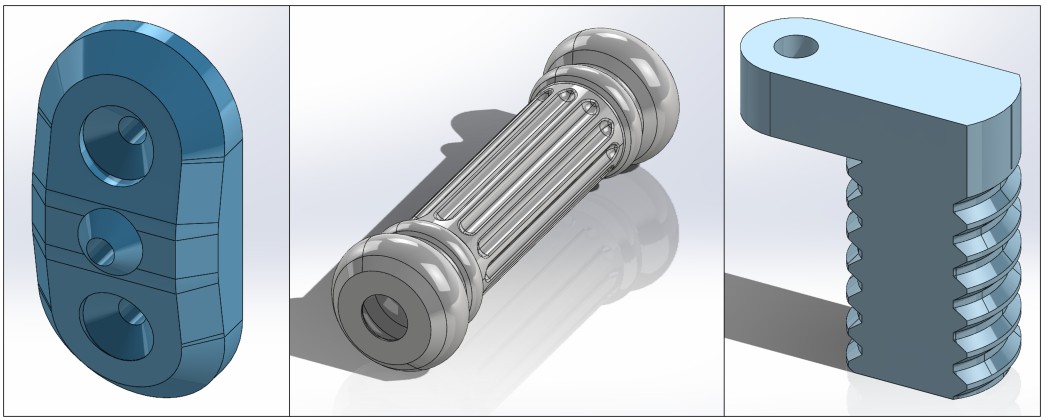

*Figure 8.* Examples of complex shapes from the Thang3D Dataset.

