# OpenReview forum: "Position: You Can't Manufacture a NeRF"
_ICML.cc/2025/Position_Paper_Track — ICML 2025 Position Paper Track poster_

### Official Review · Reviewer_pvw2 · 2025-03-05

**Significance:** 3
**Argument Clarity:** 3
**Rating:** 4
**Confidence:** 4

**Questions:**

At present, I believe the paper is worth accepting, as I also plan to research CAD generation in the near future. I would like the authors to analyze whether it is reasonable to have an MLLM generate a 3D asset by providing parameters. It should also be noted that although I have a strong background in 3D generation, I am not particularly well-versed in CAD. I merely believe that the paper indeed explores a pain point that has been largely overlooked in the academic community. I will adjust my final opinion based on the judgments of other reviewers and the authors' rebuttal.

**Discussion Potential:**

3

**Paper Summary:**

This paper examines the gap between current 3D generative models and real-world manufacturing requirements. It shows that while many models can generate 3D assets for virtual use, they often fail to account for manufacturing constraints like milling, casting, or injection molding. The study reveals that these models do not generalize well to real, human-created objects, largely due to a lack of manufacturing-specific semantics, challenges in decomposing complex shapes, and inadequate evaluation metrics. The authors propose improved datasets and benchmarks as steps toward developing models that are better suited for manufacturing applications.

**Position:**

Yes

**Position In Title:**

Yes

**Related Work:**

3

**Strengths And Weaknesses:**

First, regarding the strengths, the paper’s investigation into CAD generation is indeed addressing an important unresolved issue in the field of 3D generation. Its analysis of current methods and the introduction of a new CAD benchmark are quite reasonable. I believe that as a position paper, it does not have any major issues.

Next, the weaknesses I perceive: Since this is a task-driven study and the authors have already mentioned that current 3D generation mainly focuses on creating virtual 3D assets, I suggest that the paper should reference related methods, such as text-to-3D and single-image-to-3D. Consider citing the following papers:

[1] Dreamfusion: Text-to-3D Using 2D Diffusion \
[2] Prolificdreamer: High-Fidelity and Diverse Text-to-3D Generation with Variational Score Distillation \
[3] Zero-1-to-3: Zero-Shot One Image to 3D Object \
[4] IPDreamer: Appearance-Controllable 3D Object Generation with Complex Image Prompts \
[5] StyleTex: Style Image-Guided Texture Generation for 3D Models \
[6] Phidias: A Generative Model for Creating 3D Content from Text, Image, and 3D Conditions with Reference-Augmented Diffusion

**Support:**

3

---

> ### Author Rebuttal · Authors · 2025-04-01
>
> We thank the reviewer for their insightful comments and feedback!
>
> **Overall Comments**
>
> **Why NeRF?**
>
> [1] Dreamfusion: Text-to-3D Using 2D Diffusion
>
> [2] Prolificdreamer: High-Fidelity and Diverse Text-to-3D Generation with Variational Score Distillation
>
> [3] Zero-1-to-3: Zero-Shot One Image to 3D Object
>
> [4] IPDreamer: Appearance-Controllable 3D Object Generation with Complex Image Prompts
>
> [5] StyleTex: Style Image-Guided Texture Generation for 3D Models
>
> [6] Phidias: A Generative Model for Creating 3D Content from Text, Image, and 3D Conditions with Reference-Augmented Diffusion
>
> We can definitely add those citations to the paper, we are currently trying to fit everything in. That’s an excellent point and those papers are the primary reason we evaluated using NeRF reconstruction in the first place. Clarification of the entire experimental setup as well as clearer reasoning around why we chose NeRF will definitely be added to the paper. These papers, alongside commercial software Meshy, Spline, 3dfy, etc. are built off similar text/image-to-3D pipelines.
> Generation based on NeRF-like algorithms is one of if not the most popular mechanisms for 3D shape generation in the larger ML community, and has received much attention. These papers largely base their generative models on initially inputting or generating a set of  images, then reconstructing 3D shapes from the image set via a neural radiance field or some variant of it. This method has gained much traction for digital rendering in 3D. As discussed in reviewer oGqW’s response, we found that these methods do not generate things in a manner that is suitable for surface reconstruction. Also as stated in response to reviewer oGqW, our aim with using the original, vanilla NeRF framework to reconstruct shapes from perfect conditions was to demonstrate that even if the original images were perfect and there was an infinite number of prompt images, the remeshing process to get a surface (via an improved marching cubes algorithm) still often failed to generate a manifold surface, often creating stray edges and faces or holey meshes. This demonstrates that even if the above papers were perfect in their execution of the generation of images and subsequent creation of neural radiance fields from said images, it would still not be enough to create shapes that could be manufactured in the real world, hence the approach done by the papers above is innately flawed for the creation of surfaces. Even if one were able to achieve a meshed surface, the challenge of moving from mesh to a boundary representation for both manufacturing fidelity’s sake (how many triangles would be required to obtain a surface finish of 125 Ra – roughness average in microinches, i.e. equivalent to 3.175 microns roughness – from a mesh of a curved surface) as well as conventional feature-based machining remains.
>
> **Analyze whether it is reasonable to have an MLLM generate a 3D asset by providing parameters.**
>
> We are currently fighting the page limit here but could definitely add a small section on this. This is an excellent direction for future work but extremely challenging – namely because designs should be informed by physics (physics-based neural networks or finite element analysis approximators might be used) and are also not necessarily describable with acceptable precision in vernacular language (the domain specific languages, like what is used in BREP referenced in Figure 2, are acceptable, but people would not describe many manufactured shapes with adequate precision in spoken, colloquial English or another language). This is touched upon in the description of Figure 1 in the paper where it is asked how one would describe the shape with adequate precision for it to be generated, as that is a rather unconventional shape. Thus, for mass manufacturing, it is likely going to require a set of input parameters wildly different from what is seen in CAD-MLLM (discussed in the paper) where you merely have an image/images or pointcloud paired with a colloquial language text descriptor.

---

### Official Review · Reviewer_gdjb · 2025-03-12

**Significance:** 2
**Argument Clarity:** 1
**Rating:** 1
**Confidence:** 5

**Questions:**

1, Isn't articulation generation an important attribute?
2. How about the work on program synthesis to create CAD models?
3. There are USD data structure. Will that be useful?
4. What about material generation, texture generation etc?

**Discussion Potential:**

2

**Paper Summary:**

Describing a 3D shape in language is a difficult task as language often does not convey important shape attributes. This position paper tries to show insights how much this is important for manufacturing process and evaluated exising 3D generation works. It highlighted limitations of SOTA primitive to CAD, dataset availability, different form of representations etc.

**Position:**

Yes

**Position In Title:**

Yes

**Related Work:**

2

**Strengths And Weaknesses:**

The paper has highlighted an important topic which is very relevant today.
The paper shows the limitation of today's technology and the gaps very well.

However the paper has some major weaknesses
1. After the nice introduction, the paper divert the reader too much on shape representation.  The class of object section is not very clear.
2. Many of the paragraphs need image illustrations like line 230-235, Section 3 imprecise noise,  etc. This is one of the major drawbacks for the reader to understand almost many of the sections. Captions are not self sufficient like Figure 4.
3. References are not mentioned properly. The dataset doenot have reference in many places
4.Line 393-400 needs more elaboration and images for a ML reader.
5. The ML community takeaway is the need of 3D generation for manufacturing, which is known but there are different genre of 3D generation which is not discussed creating buckets.

**Support:**

1

---

> ### Author Rebuttal · Authors · 2025-04-01
>
> We thank the reviewer for their insightful comments and feedback!
>
> **Overall Comments**
>
> As stated in the response to reviewer ewDX (refer to overall comments for reviewer ewDX), we can generally clarify domain specific terminology better as well as add more image examples of specific kinds of shapes and failures in the appendix. We will also add more clarity on captions in general and expand the segmentation failure figure, as well as add example images of failures in the appendix.
>
> **Discussion of Shape Representation**
>
> We understand that the reviewer is concerned with the large amount of discussion on shape representation; however, shape representation is a critical piece of the issue of generative design for manufacturing – some representations of shapes (especially point clouds and meshes) are simply not suitable for mass manufacturing, and part of this paper’s purpose is to demonstrate why. An in-depth understanding of why the industry uses boundary representation for CAD/CAM and not discrete forms is critical to the argument that current generative methods (ex: NeRF) are insufficient for manufacturable design.
>
> **Dataset and Dataset References**
>
> The description of the datasets can be found in lines 159 (second column) through 171 (first column), but we can definitely clarify why in particular those datasets are significant and add images of complex shapes within each of the new datasets (see also comments to reviewer oGqW). Additionally, excluding our own unique datasets that we created specifically for this task, all the prior existing datasets (including the one we built one of our unique datasets off of) are properly referenced. Our datasets, the modified Fusion360 Segmentation dataset and the Thang3D dataset, are newly introduced in this paper and thus do not have references.
>
> **The ML community takeaway is the need of 3D generation for manufacturing, which is known but there are different genre of 3D generation which is not discussed creating buckets.**
>
> Could the reviewer please clarify the meaning behind the term “buckets”?  We think the statement implies that we should discuss more types of 3D generation.
>
> The goal of the paper is to identify the use of AI for generating shapes specifically for manufacturing in the real world, which hold a unique set of precision and accuracy requirements that are much more stringent than what is necessary for digitally rendering 3D shapes, for example for scene generation. We recognize that techniques used to generate objects in the digital rendering space are prevalent in the AI-ML community; we are trying to highlight in this position paper that the generation of 3D objects for real world construction requires a different approach. We will clarify these points in the paper and answer the following questions.
>
> **Questions**
>
>  **Isn't articulation generation an important attribute?**
>
> Articulation generation is important in rendering motion/changing movement of the object on a screen, e.g. in a scene. It is not important and not needed in specifying objects for mass manufacturing processes; instead static geometric precision to meet engineering requirements and mating shape to CNC constraints are needed. Precision manufacturing is a different and distinct application space from scene generation and rendering.
>
> **How about the work on program synthesis to create CAD models?**
>
> We can definitely add more information about this approach beyond what was already in the paper. Some of these works are discussed in section 1.4 (the works of MFCAD++ and most recently CAD-MLLM) but both of their training datasets suffer from the same core issue of being quite simplified/restricted compared to real-world data. This is elaborated on in detail in section 1.4.
>
> **There are USD data structure. Will that be useful?**
>
> As with question 1, universal scene descriptors (USD) are used for large-scale scene definition as well as motion with the scene, not our use case which is specifically to create shapes whose purpose is real-world creation of a singular object, not a scene or digital rendering of an object.
>
> **What about material generation, texture generation etc?**
>
> Material and texture are terminology used in the digital rendering world. In CNC machining, surface finish is the closest conceptual parameter to “texture”. However, in CNC manufacturing, the geometric and surface finish requirements are decoupled from the material specification – they are not ‘generated’ along with the shape as they are in scene rendering. Concepts of color and texture are available in CAD tools to visualize a design, i.e., as a means to convey the CAD to the human user, not as input to the CNC machine.

---

> > ### Comment · Reviewer_gdjb · 2025-04-05
> >
> > Thanks authors for the responses.
> > There were few concerns which this paper should address to make the reader more engaged with the topic and those concerns are partially addressed in text as an update of the paper. Like, "Many of the paragraphs need image illustrations like line 230-235, Section 3 imprecise noise, etc. "
> > Authors say "Precision manufacturing is a different and distinct application space from scene generation and rendering" in their response but in the ICML community without image based examples, it is hard to consume the differences.
> > By articulation, I was meaning parts by parts generation followed by composition so that the object can be manipulatable.
> >
> > This paper needs refinement of the claims and the limitations with illustrations using images in the context of existing generative methods.
> > Hence keeping the original score unless the points are clear for the community consumption.

---

> > > ### Author Response · Authors · 2025-04-06
> > >
> > > Hello! We thank the reviewer for their continued engagement with the paper and appreciate the latest feedback!
> > >
> > > As stated in the previous response (as well as the response to reviewer ewDX), we will add more images to the paper to clarify specific reconstruction issues as well as unique kinds of shapes. Unfortunately, we cannot add images in these rebuttal comments (due to restrictions on formatting for the comments in particular) but we will add them to the paper.
> > >
> > > "Precision manufacturing is a different and distinct application space from scene generation and rendering" is a statement specifically regarding images and rendering. We are not trying to generate images or NeRFs, for manufacturing we must use boundary representations (called .BREP files) of a shape because those files are what is required as input to manufacture a shape in the real world.  The .BREP shape file format contains mathematical definitions of a shape. The objective of this paper is to show that it is comparatively easier to generate shapes that look good on the screen (for example, images and NeRFs for digital rendering) than it is to generate .BREP files or other mathematical definitions of a shape. One cannot CNC manufacture parts directly from images (or from neural radiance fields) -- the shapes themselves must be translated into those mathematical representations that define the surface (or else, directly generate those equations as opposed to generating images or NeRFs). We need to generate in a format of an object that is permissible for manufacturing. The challenge lies in generating shapes that are in .BREP shape format, not in generating images or neural radiance fields which are not useful for manufacturing in their raw form.
> > >
> > > **ICML community without image based examples, it is hard to consume the differences.**
> > >
> > > Thus, it is relatively simpler to make images that look good, it is hard to generate .BREP files that are valid files for manufacturing. Adding generated images will not show the problem specifically because the problem is not in generating images (which the ML community is comparatively very good at), the problem is how we generate shape files that are useful for manufacturing (that is, generating .BREP files). We will add more examples of poorly generated shape files, but the generation of the file itself is the challenge -- not the image or digital rendering of the file. The issue lies in generating valid boundary representations of shapes or more generally, generating a representation of a shape that is suitable for CNC/mass manufacturing.
> > >
> > > **By articulation, I was meaning parts by parts generation followed by composition so that the object can be manipulatable.**
> > >
> > > The assembly of parts after the individual static parts have been generated is also a very interesting task but somewhat separate from what this paper tries to tackle. We can include a small section on that as a motivator for future work after this paper, but the goal of this paper is to show why we must generate shapes in other formats besides the conventional ones for digital rendering (images and NeRFs) so that they are manufacturable.

---

### Official Review · Reviewer_ewDX · 2025-03-13

**Significance:** 4
**Argument Clarity:** 2
**Rating:** 3
**Confidence:** 2

**Questions:**

* What is BREP-level accuracy, and why is it important to achieve it? It would be helpful if the authors could define relevant domain specific terms (e.g., BREP and STEP)

* Please also define what it means a shape is represented using parametric and continuous-boundary representations

* This may be a naive question, but why does post-processing meshes (e.g., making them watertight in a platform such as MeshLab) not sufficiently address limitations highlighted? In general, how are meshes “fixed” now?

**Discussion Potential:**

3

**Paper Summary:**

The paper discusses manufacturing challenges of shapes created using generative AI. To be used in physical environments, created objects need to be represented using parametric and continuous-boundary representations.

There are two ways to achieve this: transform outputs that are represented in terms of discrete primitives into a suitable representation, or directly generate shapes in the suitable representation. Common examples and limitations of these so called mesh generation (noisy outputs, insufficiently good re-meshing outputs such as incorrect normals or non-manifold edges) and primitive-to-CAD (limited by incorrect primitive segmentation of the input mesh) techniques are discussed in Section 2.1 and Section 2.2.

**Position:**

Yes

**Position In Title:**

Yes

**Related Work:**

3

**Strengths And Weaknesses:**

Strengths

The paper addresses an important limitation of existing genAI methods: how to make them suitable for subsequently creating physical shapes. The background that this paper provides is intended to provide a useful overview and position.

Weaknesses

While generally well-written, the manuscript includes few examples and specialized terminology, making it somewhat difficult to follow for a novice (particularly Section 3, 4 and 5).

It is also difficult to understand the example of the Primitive-to-CAD primitive segmentation failure. Please consider expanding Figure 5 to clarify the issue.

The experimental setup for evaluating NeRF on F360 is not described, therefore, it is difficult to follow what exactly is reported in Table 1.

Finally, in my opinion, the key contribution of the manuscript is the highlighted “need for datasets that encompass grater geometric and manufacturing complexity”, and a modification of the F360 dataset with appropriate labels, however, I did not find a description of this dataset. I would encourage the authors to detail the types of modifications needed so that they can be added to future 3D datasets to be released.

__Post Rebuttal Comments__
The authors have clarified my concerns, and I have updated my rating accordingly. This paper would be a useful contribution to the conference, highlighting the challenges that exist at the intersection of AI shape generation and shape fabrication.

**Support:**

3

---

> ### Author Rebuttal · Authors · 2025-04-01
>
> We thank the reviewer for their insightful comments and feedback!
>
> **Overall Comments**
>
> We can generally clarify domain specific terminology better as well as add more image examples of specific kinds of shapes and failures in the appendix. We will also add multiple figures in the appendix showing different examples of complex geometries vs. simplified ones. Referring to the experimental setup for the NeRF experiment, we will definitely add information about the setup and remeshing processing used (for more information, see the response to reviewer oGqW and pvw2). We will also add more clarity on captions in general and expand the segmentation failure figure, as well as add example images of failures in the appendix. More in-depth discussion of how class-based segmentation fails will be added (see also Human vs. Computer segmentation section in response to reviewer oGqW). The description of the datasets can be found in lines 159 (second column) through 171 (first column), but we can definitely elaborate the description further and also clarify why in particular those datasets are significant and add images of complex shapes within each of the new datasets. Altogether, we need more images with clearer captions, which can be added in the appendix.
>
> **General Questions**
>
> **Please also define what it means a shape is represented using parametric and continuous-boundary representations.**
>
> A continuous boundary representation is defined parametrically in a way that translates to a continuous mathematical equation, while mesh files (e.g., STL, some OBJs)  are at best an approximation of that shape using triangles. The difference is analogous to defining a circle by its true equation  (e.g., x^2 + y^2 = 4) vs attempting to approximate that circle with straight line segments.
> Considering the tolerances listed in table 5 for machining a shape in the real world, imagine how many straight line segments would be required to approximate cutting a circle with a maximum error of 1 mil = 25 microns. The number of discrete segments would be very large. For mass manufacturing, usually continuous equation-level ‘perfect’ accuracy is expected from CAD – errors will inevitably occur in the manufacturing process, approximating a shape with straight line segments and triangles to begin with would add an even greater level of error.
>
> **What is BREP-level accuracy, and why is it important to achieve it?**
>
> We can definitely clarify this in the paper. Referring to lines 93-100  as well as figure 2,  a .BREP file is a file that contains high-level, continuous closed-form equations to define the shape – this is what is referred to as a parametric, continuous-boundary representation of a shape. (STEP format files can also represent objects in this way.) The equations are sometimes wrapped in ‘objects,’ like wrapping the equation of a circle as simply a CIRCLE object that takes a center point and radius. The example shown in figure 2 is a BREP file that contains a ‘circle’ object with a defined radius and references back to another object that is the center of the circle. These mathematical definitions are perfect, with no approximations involved, and can be turned into a discrete form with arbitrary levels of precision, such as a tooling path for a CNC milling machine.
>
> **Why does post-processing meshes (e.g., making them watertight in a platform such as MeshLab) not sufficiently address limitations highlighted?**
>
> As stated in lines 187-195 in the second column, a mesh can be manifold and watertight but still a poor approximation for an object with curves as it is essentially a set of straight line segments that are connected. They would require a large number of faces to approximate it to the precision required for high-fidelity machining, which is why meshes are generally only suitable for very coarse-grain 3D printing. For manufacturing the preference is to have the curves defined mathematically as a continuous equation, which is why we use .BREP or .STEP files (as described above).
>
> **In general, how are meshes “fixed” now?**
>
> Post-processing today is largely done by humans who look at meshes and custom-select tolerances and localities for merging or removing close edges, vertices, and faces, and patching holes in the triangulation. An overview of the many techniques used is referenced in the paper as Campen et al. 2012. The techniques used are dependent on the density of the triangle mesh as well as how fine the features are – there is no fully automated mechanism that will work for any mesh in any state, there’s eventually some level of human involvement – though this potentially could be another AI/ML-suited task as a practical research challenge. For complex meshes with many features, this can be a very time-intensive manual process. We can add a short sentence or two in the paper to make these points.

---

### Official Review · Reviewer_oGqW · 2025-03-23

**Significance:** 4
**Argument Clarity:** 3
**Rating:** 5
**Confidence:** 4

**Questions:**

See above

**Discussion Potential:**

4

**Paper Summary:**

This manuscript argues that current 3D generative models are not suited for the complex geometric requirements of most manufacturing settings, instead overfitting to simple geometries that are not representative of real-world needs. The authors support this argument by showing how different generative methods fail to satisfy the tolerances required in industrial settings, as well as to segment output shapes into fabricable patches.

**Position:**

Yes

**Position In Title:**

Yes

**Related Work:**

3

**Strengths And Weaknesses:**

I congratulate the authors on their submission: I enjoyed reading it, and think it will lead to a productive discussion in the community. Overall, the position is well and thoroughly defended: it is hard for me to imagine a reader not agreeing with the main thesis of this manuscript after reading it carefully. In particular, the main strengths of the paper are its extensive evaluation of its claims, through several datasets, architectures, metrics and tasks. I particularly appreciated the enumerated list of open challenges as well as the (honest, respectful) discussion of alternative views, which I believe greatly enrich the work.

One of this manuscripts' harder jobs is to bridge the gap between the industrial geometric modeling and machine learning communities. The authors do an adequate job; however, as a geometry processing researcher, there are several points in the paper that I found at best unclear and at worst incorrect. I believe the impact of this manuscript would be much larger in my own community if the authors would address these:

- The definition of "2-manifold watertight" is unclear and contradictory at times. Sec. 1 says "meaning that if each individual shape was represented as a polygonal mesh, every edge would be incident to exactly two faces". What about self-intersecting meshes? What about non-manifold shapes that are coincident at a single point, instead of a mesh? I believe the paper may be conflating the concept of an (analytical) 2-manifold, watertight shape, and a 2-manifold *mesh*. Similarly, "in a small locality anywhere on the mesh, the mesh is guaranteed to be planar", what does this mean? A "small locality" of a mesh vertex is definitely non-planar, regardless of how small it is. "Any holes" should clarify it refers to holes in the triangulation, not holes in the shape (e.g. a torus). Finally, self-intersections are a property of a. shape's embedding, not of the topological space M; thus, who can a shape's manifoldness depend on them? I understand these considerations are orthogonal to the paper's position, but the work should be mathematically precise if it includes this mathematical language.
- I was surprised to read "Meshes are often known as 'triangle soup'", and would appreciate a citation. In my community (see, e.g., [1]), "triangle soup" and "mesh" are different terms: one is just a collection of triangles without any connectivity information, the other includes the topological adjacency information of triangles to each other.
- I was confused by the NeRF Mesh Generation results. A NeRF is not a mesh, it is an implicit representation of an object. How is a mesh extracted from this? Mathematically, this implicit representation always defines an "inside" and an "outside", so mathematically, this representation does define a manifold, non-self-intersecting boundary. What exactly does it mean to say that "50% of these shapes contained non-manifold edges"? Assuming that a mesh is extracted from a voxel grid, it is not a fair comparison to say compare the volumetric resolution of a hypothetical total grid (256^3) if only some of these grid cells result in mesh faces; instead, the comparison should be in the number of final generated faces.
- In general, I found this Mesh Generation experiment to be the main weakness in the evolution of the authors' claim. I am relatively sure that the authors' main claim would still be justified if they had done a more thorough test, but NeRFs are notorious for their lack of surface quality. Why not use SDF reconstructions, which will preserves curvature better (e.g. [3])? Or test the method against classing, non-ML reconstruction algorithms (e.g., [2])?

Some smaller comments:
- I was surprised by the discussion of the difference between how humans segment a shape vs how an algorithm does. This seems philosophical/psychological to me, and something that should either be phrased more carefully or justified with user studies / citations. It reminds me of the debate between reasoning and stochastic parrots in regards to LLMs, and I don't believe that is a settled one at all.
- Somewhere in the paper (perhaps the conclusion?), it would have been good to acknowledge that even if geometric accuracy is achieved, manufacturing systems also place other constraints on the produced object (e.g., physical, thermal, CNC reachability) that should be included in any generative process. I think this could help strengthen the author's point even more.
- I realize the authors are fighting against a page limit, so I understand it, but the discussion of "complexity" of different geometries and datasets would have really benefited from a figure showing examples of the different classes of shapes.

[1]Barill, Gavin, et al. "Fast winding numbers for soups and clouds." ACM Transactions on Graphics (TOG) 37.4 (2018): 1-12.
[2] Li, Yuanqi, et al. "Surface and edge detection for primitive fitting of point clouds." ACM SIGGRAPH 2023 conference proceedings. 2023.
[3] Wang, Zixiong, et al. "Neural-singular-hessian: Implicit neural representation of unoriented point clouds by enforcing singular hessian." ACM Transactions on Graphics (TOG) 42.6 (2023): 1-14.

**Support:**

3

---

> ### Author Rebuttal · Authors · 2025-04-01
>
> We thank the reviewer for their insightful comments and feedback!
>
> **Definition of 2-manifold watertight mesh and ‘locally planar’**
>
> We will clarify this and state all assumptions in the paper, this is an excellent point. We assume that the shapes are to be created in the real world and therefore their representative meshes when embedded in 3-space must be injective in that space.
>
> By ‘locally planar’ (which was a poor choice in simplified phrasing that will be amended), we mean a 2-manifold mesh is a discrete approximation of a smooth surface where every vertex's local neighborhood—often called its "star"—is topologically equivalent to an open disk (and there are no half-disks since there are no boundaries – or there shouldn’t be in what we call valid meshes). Just as each point on a continuous 2-manifold locally resembles the Euclidean plane, each vertex in a manifold mesh must have incident faces arranged in a single, connected, cyclic order without gaps or overlaps.
>
> Further clarifying ‘watertight’ specifically for our application is challenging, but the papers referenced by our paper are also referenced in an article which provides a succinct and well-supported definition of a valid mesh, based on definitions provided by Botch,, Edelsbrunner, and Giblin. In strict geometric terms:
> 1. A self intersection is an intersection of two faces of the same mesh.
> 2. A non-manifold edge does not have exactly 2 incident faces.
> 3. The star of a vertex is the union of all its incident faces.
> 4. A non-manifold vertex is a vertex where the corresponding star is not connected when removing the vertex.
> 5. A mesh is 2-manifold if it does contain neither self intersections, nor non-manifold edges, nor non-manifold vertices.
> 6. A 2-manifold mesh is called watertight if each edge has exactly two incident faces, i.e. no boundary edges exist.
> Mario Botsch, et Al. Polygon Mesh Processing.
> H. Edelsbrunner. Surface Reconstruction by Wrapping Finite Sets in Space.
> P. Giblin. Graphs, Surfaces and Homology. Cambridge University Press
>
> **Meshes & ‘triangle soup’ – Manufacturing Community ‘meshes’ vs geometry meshes**
>
> That’s a great point, the phrasing on this statement should actually be reversed (and will be amended in the paper) in that triangle soup (STL files) are considered meshes in the manufacturing community and software, but by graphics standards STL files are triangle soup since they contain triangles defined solely by vertices without connectivity information. As shown in figure 3, STLs are a collection of vertices associated with a particular face. Major CAD software will ‘save’ or ‘load’ a ‘mesh’ from STL or OBJ format, which is why in manufacturing and 3D printing communities these files are called ‘meshes’ even though they do not enforce the mesh criteria.
> See Hayong Shin, et Al., "Efficient topology construction from triangle soup," Geometric Modeling and Processing.
>
> **NeRF mesh generation**
>
> NeRFs do not directly produce surfaces—a remeshing algorithm was used to generate a mesh from surface points, and we base our metrics on the resultant mesh (final generated faces). As to why we picked NeRF, it is extremely popular in the rendering community and we felt it important to address them in this position paper as many, including reviewer pvw2, would like to know the feasibility of using their approach to generate manufacturable shapes. See also response to reviewer pvw2.
>
> **Why not SDF or non-ML reconstruction algorithms?**
>
> While analytical methods might work for reconstructing shapes (e.g., RANSAC-based reconstruction), we are unaware of any analytical generation algorithms besides conventional topology optimization—and SDFs are also not as prevalent. Our goal with the NeRF experiment was to show that popular NeRF-based processes cannot even perform simple reconstruction well from perfect inputs, and therefore cannot achieve the required fidelity for manufacturing.
>
> **Human vs. Computer Segmentation**
>
> The discussion of what is more human can definitely be removed from the paper as ‘human’ is a bit of a subjective term; however, what can be objectively stated is that humans do have the ability to segment things without having preset primitives for each segment. A major issue with current ML (and analytical RANSAC-based) approaches is segmentation is performed on predefined shape categories rather than by raw physical characteristics. If a segment does not fall into a defined category—even with a drastic change in geometry—category-based segmentation algorithms may fail to split it. As seen in figure 1, there are many logical places to draw boundaries based on changes in curvature, yet the segments might not fit into categories like ‘plane’ or ‘cylinder.’
>
> **Accurate geometry and manufacturing constraints**
>
> This is an excellent point and could be a position paper entirely on its own. We can add a small section on this.
>
> **Complexity of different geometries and the new dataset**
>
> See response to reviewer ewDX.

---

> > ### Comment · Reviewer_oGqW · 2025-04-04
> >
> > Thank you for your answers! I was already positive about this work, and your responses make even more positive. I will raise my score accordingly.
> >
> > I was really surprised to see the negative reviews. After reading them, I see that some reviewers seem to have mostly misunderstood the paper or be unfamiliar with shape representations and the importance of the problem (or rather even with what the problem is). I encourage the authors to not be frustrated by this and instead treat it as a data-point on how their current introduction and beyond can be hard for a general ML audience to understand, and edit it accordingly and patiently. That being said, **I hope the AC takes this lack of understanding/familiarity into account when weighing the different reviews and deciding whether to accept this work**, which I strongly believe should make it to the conference.

---

### Decision · Program_Chairs · 2025-04-30

**Decision:**

Accept (poster)

**Comment:**

This paper received divergent reviews. After rebuttal, all but one reviewers converged on acceptance. Reviewers agree that this paper presents an important concern, and that it is well argued and well written.

The one remaining negative concern relates to the absence of visual examples. The authors argue that the important limitations are not visual, but rather have to do with the underlying representation. While the authors are correct on this point, it is useful for an ML audience which is not familiar with the complexities of shapes and their representations to see the data distribution and the kinds of representations that is needed. For this purpose, I encourage the authors to include in the appendix of more renderings of shapes from the evaluation dataset, and including more BREP / STEP files along with the corresponding shapes.

The lack of visualizations is a fairly minor comment, so I am recommending acceptance.

Authors should consider revising/expanding the title to make it more accessible to a broad audience. Otherwise, those unfamiliar with the acronyms may skip this paper.